# Fully Automated Production of (((*S*)-1-Carboxy-5-(6-([^18^F]fluoro)-2-methoxynicotinamido)pentyl)carbamoyl)-l-glutamic Acid ([^18^F]JK-PSMA-7) [note 1]

**DOI:** 10.3390/ph18010119

**Published:** 2025-01-17

**Authors:** Philipp Krapf, Thomas Wicher, Boris D. Zlatopolskiy, Johannes Ermert, Bernd Neumaier

**Affiliations:** 1Forschungszentrum Jülich GmbH, Institute of Neuroscience and Medicine, Nuclear Chemistry (INM-5), Wilhelm-Johnen-Str., 52428 Jülich, Germany; p.krapf@fz-juelich.de (P.K.); j.ermert@fz-juelich.de (J.E.); 2Institute of Radiochemistry and Experimental Molecular Imaging, Faculty of Medicine and University Hospital Cologne, University of Cologne, Kerpener Str. 62, 50937 Cologne, Germany; boris.zlatopolskiy@uk-koeln.de (B.D.Z.)

**Keywords:** [^18^F]JK-PSMA-7, positron emission tomography (PET), prostate cancer imaging, radiofluorination, Trasis AllInOne (AIO) module, automation, GMP

## Abstract

**Background:** The radiotracer [^18^F]JK-PSMA-7, a prostate cancer imaging agent for positron emission tomography (PET), was previously synthesized by indirect radiofluorination using an ^18^F-labeled active ester as a prosthetic group, which had to be isolated and purified before it could be linked to the pharmacologically active Lys-urea-Glu motif. Although this procedure could be automated on two-reactor modules like the GE TRACERLab FX2N (FXN) to afford the tracer in modest radiochemical yields (RCY) of 18–25%, it is unsuitable for cassette-based systems with a single reactor. **Methods:** To simplify implementation on an automated synthesis module, the radiosynthesis of [^18^F]JK-PSMA-7 was devised as a one-pot, two-step reaction. The new method is based on direct (“late-stage”) radiofluorination of an appropriate onium triflate precursor and subsequent deprotection with *ortho*-phosphoric acid. It was successfully established on the cassette-based Trasis AllInOne (AIO) module. **Results:** Overall, the new protocol enabled the production of [^18^F]JK-PSMA-7 in activity yields of 39 ± 4% (RCY = 58%) with an overall synthesis time of about 1 h. In a single production run with an initial activity of 36-43 GBq, 13-19 GBq of [^18^F]JK-PSMA-7 with a radiochemical purity of >99% was obtained. **Conclusions:** We have established a highly reliable, GMP-compliant process for the automated radiosynthesis of [^18^F]JK-PSMA-7 on the Trasis AllinOne (AIO) synthesizer, ensuring consistent and efficient production of this radioligand.

## 1. Introduction

Prostate cancer (PCa) is the second most common malignancy and the fifth most common cause of cancer-related death in men [1]. Early diagnosis and accurate staging of the disease are critical for optimizing treatment strategies and improving patient outcomes. Among the most promising targets for PCa imaging and therapy is the prostate-specific membrane antigen (PSMA), a transmembrane enzyme highly overexpressed on prostate cancer cells [2]. Small molecule PSMA inhibitors based on a Lys-urea-Glu motif exhibit high and selective uptake by PCa cells, which has been exploited for diagnostic and therapeutic applications by coupling them to different radionuclides [3,4].

In nuclear medicine, two imaging modalities are routinely used for PCa diagnostics: single photon emission computed tomography (SPECT) and positron emission tomography (PET). Although SPECT is more widely available, PET offers a superior spatial resolution, making it the preferred choice for PCa imaging [5]. Over the past decade, several PSMA-targeting PET tracers have been successfully introduced into clinical practice, revolutionizing PCa diagnostics [3,4,6,7,8,9]. PSMA-PET has largely supplanted other PET modalities, such as choline PET [10], due to its superior diagnostic performance, particularly in recurrent disease with low prostate-specific antigen (PSA) values [11].

Despite these advancements, clinical studies have reported a growing number of false-negative findings with existing PSMA ligands [12]. The initial development of PSMA-PET radioligands focused on radiometal-based tracers, such as ^68^Ga-labeled compounds, which remain pivotal tools for PCa diagnostics [6,13]. The coupling of the Lys-urea-Glu motif with a chelator also allowed for radiolabeling with therapeutic radiometals, such as lutetium-177 [14] or actinum-225 [15], paving the way for the development of theranostic pairs. However, the increasing demand for PSMA-PET imaging has highlighted the production limitations of ^68^Ga-based tracers, necessitating the development of alternative radiolabeled PSMA ligands, particularly those based on fluorine-18 [16].

In this regard, the development of [^18^F]DCFPyl [17] was a major breakthrough that spurred the development of improved ^18^F-labeled radioligands based on the Lys-urea-Glu motif [18,19,20,21].

One of these ^18^F-labeled derivatives is [^18^F]JK-PSMA-7 ((((*S*)-1-carboxy-5-(6-[^18^F]fluoro-2-methoxynicotinamido)pentyl)carbamoyl)-l-glutamic acid), which showed more favorable properties for tumor localization after biochemical recurrence of prostate cancer [22,23,24,25]. The synthesis of [^18^F]JK-PSMA-7 was originally performed by indirect radiofluorination with an ^18^F-labeled active ester as prosthetic group [22]. The latter was prepared by a classical nucleophilic aromatic substitution (S_N_Ar) reaction of a trimethylammonium leaving group in the ortho position on the activated pyridine ring of tetrafluorophenyl (Tfp) ester **1** (Figure 1). The resulting ^18^F-labeled building block (6-[^18^F]fluoro-2-methoxypyridin-3-yl)-(2,3,5,6-tetrafluorophenyl)methanone ([^18^F]**2**) was then coupled with the pharmacophore Lys-C(O)-Glu (**3**) to afford [^18^F]JK-PSMA-7 ([^18^F]**4**) in 18–25% radiochemical yields (RCYs). To obtain sufficiently large activity amounts for medical use and to meet the regulatory requirements of “Good Manufacturing Practice” (GMP) for radiopharmaceuticals [26,27,28], automation of the production process is essential [29,30]. Although high-quality products meeting GMP standards can also be produced using conventional, manually operated systems [31], these systems introduce additional challenges for regulatory compliance, such as the need for cleaning validation according to EU GMP Annex 15: Qualification and Validation. In contrast, cassette-based automated systems minimize the risk of cross-contamination, streamline GMP compliance, and enable more efficient production of radiopharmaceuticals like [^18^F]JK-PSMA-7 ([^18^F]**4**).

However, due to the additional purification step required to isolate the ^18^F-labeled building block, the indirect two-step radiolabeling approach for the production of [^18^F]JK-PSMA-7 can be only implemented in two-reactor modules, such as the commercially available GE TRACERLab FX2N [32]. Moreover, final isolation of the product should be carried out by solid phase extraction (SPE) to avoid semipreparative HPLC purification. Therefore, the aim of the present work was to develop a simple and robust one-pot synthesis of [^18^F]JK-PSMA-7 amenable to automation by direct (“late-stage”) radiofluorination of an onium triflate precursor [33].

The prerequisite to obviate an additional purification step and enable the application of a single reactor module was the introduction of a direct radiofluorination step for the preparation of [^18^F]JK-PSMA-7 ([^18^F]**4**). Additionally, the process should be amenable to automation in a synthesizer with disposable cassettes. The application of disposable cassettes reduces the risk of cross-contamination and operating errors and is in full compliance with GMP regulations [34,35]. Accordingly, a robust and GMP-compliant process for the production of [^18^F]JK-PSMA-7 ([^18^F]**4**) was established on the versatile cassette-based Trasis AllinOne (AIO) synthesizer [36], a module well-established for clinical production of ^18^F-labeled radiopharmaceuticals [37,38].

## 2. Results and Discussion

The original synthesis of [^18^F]JK-PSMA-7 ([^18^F]**4**) was a two-step build-up synthesis that required two labeling precursors [22]. The ^18^F-labeled building block, (6-[^18^F]fluoro-2-methoxypyridin-3-yl)-(2,3,5,6-tetrafluorophenyl)methanone ([^18^F]**2**), was synthesized by nucleophilic ^18^F-fluorination of 6-methoxy-*N*,*N*,*N*-trimethyl-5-(2,3,5,6-tetrafluorobenzoyl)pyridin-2-aminium triflate (**1**) via a modified minimalist approach [39]. To this end, [^18^F]fluoride was fixed on a QMA cartridge, washed with anhydrous acetonitrile (MeCN) and eluted with a solution of precursor **1** and tetrabutylammonium bicarbonate (TBAB) in MeCN followed by a mixture of MeCN and *t*BuOH. The reaction mixture was then heated and the resulting intermediate [^18^F]**2** was purified by SPE before it was coupled to the urea derivative Lys-C(O)-Glu (**3**). To accomplish this, the active ester [^18^F]**2** was eluted into a solution of **3** and Et_4_NHCO_3_ in anhydrous EtOH and the reaction mixture was stirred for 3–5 min. After hydrolysis of the protecting groups with 0.1% TFA solution, the product was purified by SPE. To automate this two-step process using two precursors with an intermediate purification step, a device with two reactors was required. The GE TRACERLabTM FX2N synthesizer was selected for this purpose. To improve the purity of the product, a final HPLC purification had to be integrated into the process (see supporting information of [22] for details). To remove the MeCN used as HPLC solvent, the product fraction was diluted with water and fixed on a SPE cartridge. The cartridge was then washed with water, the product was eluted with EtOH in isotonic saline, and the eluate was diluted with isotonic saline to reduce the alcohol content to 10%. The entire process took about 80 min and was prone to errors due to the many intermediate cleaning steps. Furthermore, two labeling precursors used under GMP conditions, as also used in the [^18^F]PSMA-1007 synthesis [20], had to be considered for clinical application. Therefore, the process was reduced to a simplified two-step, one-pot synthesis, and a cassette-based system, the Trasis AllinOne [38], was chosen as the synthesizer.

To enable one-pot preparation of [^18^F]JK-PSMA-7 ([^18^F]**4**) by direct radiofluorination, the precursor 6-methoxy-*N*,*N*,*N*-trimethyl-5-(2,3,5,6-tetrafluorophenoxycarbonyl)pyridine-2-aminium triflate (**5**) was prepared by coupling Tfp ester **1** with a *tert*-butyl (*t*Bu)-protected analog of the pharmacophore **3** (Figure 2). The synthesis of [^18^F]JK-PSMA-7 by “late-stage” ^18^F-fluorination could then be performed via an ^18^F-for-N^+^(CH_3_)_3_ exchange reaction of **5** and deprotection of the resulting intermediate in the same reactor. To this end, [^18^F]fluoride fixed on a QMA cartridge was eluted with a solution of tetrabutylammonium hydroxide-30-hydrate (TBAOH) in MeCN/water and dried three times.

Precursor **5** (10 mg), present as a solid in the cassette, was automatically dissolved in 1 mL of MeCN immediately before the start of the synthesis. The ^18^F-for-N^+^(CH_3_)_3_ exchange could be performed within 5 min at 70 °C. Subsequent hydrolysis of the *t*Bu-protecting groups was achieved by adding 1 mL of *ortho*-phosphoric acid into the reactor and heating the reaction mixture for 5 min at 70 °C. Phosphoric acid was used due to its lower toxicity compared to other acids like, e.g., trifluoroacetic acid. After cooling and dilution of the reaction mixture with saline, the crude product solution was directly transferred to a semi-preparative HPLC system followed by SPE purification.

The total activity yield of [^18^F]JK-PSMA-7 ([^18^F]**4**) obtained using the AIO system was 38.9 ± 4.0% (*n* = 260; radiochemical yield of 58%), with an overall synthesis time of about 45 min. In a single production run with a starting activity of 36–43 GBq, 13–19 GBq of [^18^F]**4** was obtained. The radiochemical purity of the product was >99% (cf. Figure 3). No radiolysis was observed even at high radioactivity levels (>300 GBq). Compared to the previous two-reactor process, the direct synthesis consistently provided more than 20% higher activity yields (non-decay corrected). In addition, the use of a cassette-based system simplified the preparation and was advantageous with regard to GMP compliance.

The GC method described in “Section 3.2.6. Quality Control and Validation of Analytical Procedures” allowed for accurate and specific quantification of MeCN (the residual solvent) and EtOH (the excipient) in the presence of other components of the sample matrix of a sterile [^18^F]JK-PSMA-7 solution for injection. The presented results prove the suitability of the chosen GC method for the determination of MeCN and EtOH content in a [^18^F]JK-PSMA-7 solution for injection.

The HPLC method described in “Section 3.2.6. Quality Control and Validation of Analytical Procedures” enabled clear identification of JK-PSMA-7 with a retention time of 6.4 min. The validations showed that the deprotected precursor was quantitatively separated. The accuracy, expressed as percent recovery, was within the 90–110% limit at the selected concentrations. Precision was assessed by the coefficient of variation (CV) of the area under the peak, which did not exceed the predetermined value of 5% for three injections at the same concentration (9.15, 10.25 and 10.10). Linearity and range were evaluated by creating a calibration curve. The correlation coefficient R was >0.990, while the range was 1.035–12.43 μg/mL. The limit of quantification (LOQ) was set at 1 μg/mL. The retention time of [^18^F]JK-PSMA-7 was compared with the retention time of an external JK-PSMA-7 standard to prove that the method is suitable for detecting the radiochemical identity. For this purpose, a batch of [^18^F]JK-PSMA-7 and an external standard sample were analyzed using the method. The corresponding chromatograms can be found in Appendix A. The retention time of the external JK-PSMA-7 standard was 5.94 min; that of [^18^F]JK-PSMA-7 was 5.87 min. A delay time for the active detector of 0.1 min is stored in the method. The uncorrected time was, therefore, 5.97 min. The deviation from the standard thus corresponded to 0.03 min or 0.5% relative to the retention time of the external standard. The delay time was reduced to approx. 0.05 min. The requirement of a maximum deviation of 3% is fulfilled. The method is suitable for detecting radiochemical identity.

The results confirmed the suitability of the selected analytical HPLC method for the quality control of a [^18^F]JK-PSMA-7 solution for injection.

To enable detection of [^18^F]fluoride in the final solution as a radiochemical impurity, a TLC method was validated in addition to the HPLC method. However, it turned out that the [^18^F]fluoride causes two peaks. A narrow one with a retention time of approx. 0.9 min and a very broad one with a retention time of approx. 3 min. Hydrofluoric acid has a pKs value of 3.1. The eluent used (water:TFA, 1000:1) had a pH value of 1.9. Accordingly, a large proportion of the hydrofluoric acid was present as undissociated HF, which interacts strongly with the stationary phase or the silica particles. It could be shown that the [^18^F]fluoride can be detected with the method up to a concentration of 2.7 MBq/mL. However, the method is not suitable for determining the proportion of [^18^F]fluoride.

Thus, quality control tests performed on the formulated and dispensed final product solution were fully compliant with release criteria based on the specifications given in the European Pharmacopoeia for the synthesis of ^18^F-labeld radiotracers (for details, see Section 2, Table 1). The routine analytical HPLC chromatogram of the final product (Figure 3) showed a retention time of 6.5–7.0 min for [^18^F]**4**. A total of five UV impurities could be detected, all of which comply with the acceptance criterion (Figure 3). The amount of unknown UV impurities (Figure 3: UI 1–5) was below the limit of 5 µg/mL (Table 1). Tetrabutylammonium hydroxide with a detection limit of <260 μg/mL could not be detected in any sample.

As part of the validation of the synthesis, the stability of the [^18^F]JK-PSMA-7 injection solution was also examined over a period of 8 h. No significant decomposition was detected during this time frame, demonstrating that the product is stable and remains suitable for use within this period.

## 3. Materials and Methods

Unless specified otherwise, all reagents and solvents were purchased from Sigma Aldrich (Darmstadt, Germany) or VWR International (Darmstadt, Germany) and used without further purification. Saline solution for injection (0.9% NaCl *w*/*v*%) and water for injection (WfI) were acquired from B. Braun (Melsungen, Germany). Sep-Pak Accell Plus QMA carbonate and Sep-Pak C18 Plus cartridges were purchased from Waters Corporation (Eschborn, Germany). The non-radioactive reference standard (((*S*)-1-carboxy-5-(6-fluoro-2-methoxynicotinamido)pentyl)carbamoyl)-l-glutamic acid (**4**) and the radiolabeling precursor 5-(((*S*)-6-(*tert*-butoxy)-5-(3-((*S*)-1,5-di-*tert*-butoxy-1,5-dioxopentan-2-yl)ureido)-6-oxohexyl)carbamoyl)-6-methoxy-*N,N,N*-trimethylpyridin-2-ammonium triflate (**5**) were obtained from Trasis (Ans, Belgium).

### 3.1. Production of No-Carrier-Added [^18^F]Fluoride

[^18^F]Fluoride was produced using the ^18^O(p,n)^18^F reaction via bombardment of enriched [^18^O]water with 16.5 MeV protons and a beam current of about 20 µA in a titanium target at the BC1710 cyclotron (The Japan Steel Works Ltd., Shinagawa, Japan) at the INM-5 (Forschungszentrum Jülich). The typical amount of [^18^F]fluoride produced was approximately 36–43 GBq.

### 3.2. Automated Synthesis of [^18^F]JK-PSMA-7 on the Trasis AllInOne Synthesizer

The automated synthesis sequence was developed in-house in cooperation with Trasis and consisted of four main steps: trapping and release of cyclotron-produced [^18^F]fluoride using an ion-exchange cartridge, ^18^F-fluorination of the radiolabeling precursor, and purification and formulation of the final product. The sequence and the corresponding cassette were originally developed in-house with the support of Trasis (Figure 4 and Figure 5). The cassette was unwrapped and assembled no more than 24 h before the start of synthesis (SOS).

#### 3.2.1. Cassette Preparation

All reagents were prepared and organized on the same day as tracer production. The QMA eluent and precursor were prepared during cyclotron bombardment, as closely as practically feasible before the activity transfer.

A vial with tetrabutylammonium hydroxide-30-hydrate (TBAOH; 11.5 mg, 60.1 mmol) in MeCN/water (85/15 *v*/*v*%; 1.5 mL) for elution of [^18^F]fluoride from the QMA cartridge was attached to position 2 (A) of the cassette.A vial with solid labeling precursor **5** (10 mg; 12.1 µmol) was attached to position 8 (B) of the cassette.A vial with phosphoric acid (1.1 mL, 85%) was secured to position 11 (C) of the cassette.A vial with anhydrous MeCN (1 mL) was coupled to position 12 (D) of the cassette.A vial with absolute EtOH (10.35 mL) was attached to position 15 (E) of the cassette.A vial filled with 100 mg of sodium ascorbate was secured to position 16 (F) of the cassette.A storage bag with sterile saline solution (250 mL) and the final product vial were connected (via a three-way valve) to position 18 (G) of the cassette.

#### 3.2.2. HPLC Unit

The HPLC eluent A (red bottle) was prepared by adding 2 mL phosphoric acid to 500 mL WfI. HPLC eluent B (yellow bottle) consisted of 150 mL MeCN, while HPLC eluent C (green bottle) consisted of 500 mL WfI. Column: XBridge BEH Shield RP18 5 µm, 10 mm × 250 mm (Waters, Germany); flow-rate: 5 mL/min.

#### 3.2.3. Preliminary Steps Before Start of the Synthesis 

The steps are listed in Table 2.
The precursor (vial in position B) was dissolved in 1 mL of MeCN (from vial in position D).The C-18 cartridge was preconditioned with 5 mL of EtOH (from vial in position E) followed by 16 mL of saline (from storage bag in position G).Sodium ascorbate was dissolved in 10 mL of saline (from storage bag in position G) and passed into the final product vial.The sodium ascorbate syringe was rinsed with 10 mL of saline (from storage bag in position G) and the saline solution passed into the waste.A total of 12.6 mL of saline (from storage bag in position G) was drawn up with the syringe and passed into the now empty sodium ascorbate vial, which was later used to collect and dilute the HPLC product fraction of about 5 mL (see above).Finally, the cassette was dried with N_2_.

#### 3.2.4. Preparation of Dispensation

Dispensing was performed in a GMP Grade hot cell equipped with a fully automated close vial dose divider system built at FZ Jülich. The product line coming from the Trasis AIO was first rinsed with 5 mL of 70% EtOH, followed by 10 mL of water for injection into the dose calibration vessel. The product line was then dried with a stream of helium and the dose calibration vessel was replaced.

#### 3.2.5. Production of [^18^F]JK-PSMA-7

[^18^F]Fluoride was produced via irradiation (1–1.5 h) of ^18^O-enriched water with 16.5 MeV protons and a beam current of about 20 µA using the ^18^O(p,n)^18^F nuclear reaction. Irradiations were performed at the baby cyclotron BC1710 (Japan Steel Works) at the Forschungszentrum Jülich using a titanium target. After end-of-bombardment (EOB), the [^18^F]fluoride-containing (~37 GBq) [^18^O]water solution was transferred from the cyclotron directly into the collecting syringe (3 mL) of the module. The [^18^F]fluoride was fixed on a QMA cartridge while the reactor was simultaneously heated to 65 °C. The collecting syringe eluted the QMA cartridge with 0.3 mL of the eluent. Subsequently, an additional 0.8 mL of air was pumped through the cartridge with the syringe. [^18^F]Fluoride was collected in the preheated reactor, the temperature of the reactor was increased to 80 °C and the [^18^F]fluoride was subjected to three successive drying steps under a stream of N_2_ and reduced pressure (80 s at 115 °C, 180 s at 125 °C and 124 s at 95 °C). The reactor was then cooled to 55 °C, the precursor dissolved in 1 mL MeCN was added and ^18^F-fluorination was carried out at 70 °C for 5 min. After cooling to 40 °C, phosphoric acid was added in several portions and the reactor was heated to 50 °C. For hydrolysis, the temperature was kept at 50 °C for another 7 min under reduced pressure. Heating was then switched off and the second syringe was used to transfer 3 mL of saline solution from the storage bag into the reactor. The same syringe was then used to recover and mix the entire contents of the reactor and to transfer the thoroughly mixed solution into the HPLC loop. To recover any remaining product, the reactor was rinsed with another 3 mL of saline solution from the storage bag, which was also transferred into the HPLC loop. While the product was purified by HPLC, the syringe was cleaned by drawing up 18 mL of saline and dispensing it into the waste. Thereafter, 4 mL of EtOH and air were drawn up and also dispensed into the waste. After that, the syringe was rinsed two more times with saline.

The HPLC fraction containing the product was directly transferred into the sodium ascorbate vial (V16) filled with saline. The collected product solution was then aspirated (in two portions) with the syringe and fixed on the C18 cartridge. The residual solution was transferred to the waste and the C18 cartridge was washed with 12 mL of saline. The product was then eluted with 1.35 mL of EtOH and transferred to the final product vial, and the C18 cartridge was rinsed two times with saline. Finally, the product solution was transferred to the collecting vessel of the dispensing apparatus for automatic dispensing (through a sterile filter) into sterile vials.

#### 3.2.6. Quality Control and Validation of Analytical Procedures

The quality control specifications are summarized in Table 2. The product always met these specifications. After the product was transferred to the dispensing cell, a sample was taken for quality control and used to determine the appearance by visual inspection, the radionuclide identity by half-life measurements with a dose calibrator (ISOMED 2010; NUVIATech healthcare, Dresden, Germany) and the pH value with a pH-meter (SevenExcellence S400, Mettler Toledo, Gießen, Germany).

##### HPLC for Identification, Chemical and Radiochemical Purity Determination

Method validation was performed for HPLC identification and determination of chemical and radiochemical purity of the [^18^F]JK-PSMA-7 injection solution in compliance with ICH Guideline Q2(R1) Validation of Analytical Procedures and European Pharmacopoeia Chromatographic Separation Techniques 07/2016:20246.

The HPLC module consisted of the following Merck-Hitachi (Darmstadt, Germany) components:

Merck Hitachi HPLC Pumpe LaChrome Model L 2130 with degaser, Merck Hitachi HPLC Organizer Box LaChrom L2000, Merck Hitachi LaChrom Model L-2450 DAD Dioden-Array-Detektor, and Merck Hitachi HPLC Column Oven LaChrom Model L-2300. The system was coupled in series to a NaI pinhole radio flow detector and GABIstar radioactivity HPLC flow monitor for radioactivity measurements (Elysia-Raytest, Straubenhardt, Germany). The entire setup was controlled by EZChrom software, Version A.04.08 (Agilent, Waldbronn, Germany).

Analysis was performed on a Prontosil 120-5-C18 ace-EPS (125 × 4.6 mm) column at a flow rate of 1.5 mL/min using gradient elution with water/trifluoroacetic acid (1000:1 *v*/*v*%) (=A) and EtOH (=B). The gradient started at 85% A and 15% B; after 9 min, the system changed to 55% A and 45% B, and after 15 min, the gradient was reset to 85% A and 15% B. The measurement was completed after 25 min. The spectrum was recorded at 230 nm. Prior to [^18^F]JK-PSMA-7 injection, system suitability was validated via injection of a freshly prepared JK-PSMA-7 reference solution (10 μg/mL, 20 μL). The recovery of JK-PSMA-7 amounted to 100 ± 15%.

##### Validation of Chemical Impurities (UV Detector)

The following criteria were assessed during the validation: specificity, accuracy, linearity and precision. Range, detection and quantification limits were also determined. Linearity was essential for performing A_m_ calculations, using the slope and intercept of the line of best fit.

##### Validation of Radiochemical Impurities (Radiodetector)

The following criteria were assessed during the validation: accuracy, linearity, specificity and precision. Range, detection and quantification limits were also determined.

##### GC for Residual Solvent and Ethanol Content Determination

Method validation was performed for gas chromatography (GC) identification and control of residual solvents in compliance with European Pharmacopoeia guidelines of Gas Chromatography 01/2008:20228, Residual Solvents 07/2016:50400, Identification and Control of Residual Solvents 07/2017:20424 and Chromatographic Separation Techniques 07/2016:20246 as well as the ICH guideline Q2(R1) Validation of Analytical Procedures. EtOH was used as an excipient to enhance stability of the product and was not considered a residual solvent. The content of EtOH was determined to verify its compliance with a maximum volumetric concentration of 10% *v*/*v*%.

The GC module consisted of the following Agilent (Waldbronn, Germany) components:

GC model Agilent 6850 equipped with a flame ionization detector, an Agilent J&W CP-Select 624 CB GC column (25 m, 0.15 mm, 0.84 μm) and an autoinjector (Agilent 7683B Automatic Liquid Sampler). The GC analyses were performed using a 1 µL injection volume, a split ratio of 1:50 and helium as the carrier gas. The temperature was programmed to 40 °C for 4 min after injection, followed by a ramp to 180 °C at 120 °C/min, a hold at 240 °C for 6.5 min, and a cool to 35 °C. The system was controlled by EZChrom software, Version A.04.08.

##### Validation of GC Analysis of Residual Solvents

The following criteria were assessed during the validation: specificity (MeCN, EtOH), linearity of EtOH and precision.

##### Thin Layer Chromatography Tests

The determination of the tetrabutylammonium hydroxide content was carried out by means of thin-layer chromatography (TLC). For this purpose, 2 μL of sample and 2 μL of a reference solution were placed on a Silicagel 60 DC plate. A mixture of MeOH and 25% ammonia solution (90:10, *v*:*v*) was used as solvent, with a running distance of 6 cm. After drying for approx. 10 min at room temperature in air (fume cupboard), the TLC plate was placed in an iodine chamber until the spots became visible (approx. 10 min).

The radiochemical purity ([^18^F]fluoride content) was also determined using TLC. For this purpose, 2 μL of the sample was placed on a Silicagel 60 TLC plate with a running distance of 6 cm. A mixture of 90% MeOH, 7% water and 3% acetic acid (100%) was used as the solvent. After drying for approx. 5 min at room temperature in air, the TLC plate was scanned with a TLC scanner MiniGITA (Elysia-Raytest).

##### Bacterial Endotoxin Test and Sterility

Bacterial endotoxin content was analyzed by a portable Endosafe^®^ nexgen-PTS™ reader (Charles-River, Sulzfeld, Germany) according to the guidelines of the European Pharmacopoiea on Bacterial Endotoxin 2.6.14. Sterility testing was performed at Labor L&S (Bad Bocklet, Germany) using direct inoculation methods.

## 4. Conclusions

[^18^F]JK-PSMA-7 was prepared in 38.9 ± 4.0% activity yields by a one-pot synthesis which entails direct radiofluorination of a *t*Bu-protected onium triflate precursor and subsequent deprotection of the intermediate in the same reactor. The method represents a significant improvement over the indirect radiofluorination procedure that affords the tracer in lower yields and can only be performed in a two-reactor module. One of the particular merits of the direct approach is its robustness and reliability, as demonstrated by almost constant yields in 260 successful syntheses. The use of a cassette-based system enables GMP-compliant production of the tracer in any modern radiopharmaceutical laboratory in high quantities. However, the adoption of cassette-based systems assumes access to this specialized equipment, which may not be available in all facilities. Nevertheless, the increased synthesis reliability and improved GMP compatibility makes them a worthwhile choice for the production of [^18^F]JK-PSMA-7 as well as other radiopharmaceuticals. Another potential limitation of the synthesis lies in the use of the higher starting activities of fluorine-18. In this study, no radiolysis issues were observed with the 40 GBq starting activity used. Nevertheless, to mitigate potential risks, the formulation was carried out with sodium ascorbate as a radical scavenger. This precaution ensures that syntheses with starting activities of up to 100 GBq could be performed without radiolysis-related complications. However, for even higher starting activities, the risk of radiolysis cannot be entirely excluded and would require further investigation.

## Figures and Tables

**Figure 1 pharmaceuticals-18-00119-f001:**
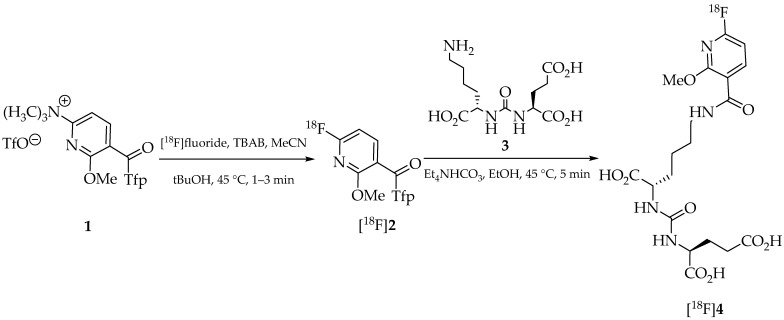
Synthesis of [^18^F]JK-PSMA-7 ([^18^F]**4**) by indirect radiofluorination.

**Figure 2 pharmaceuticals-18-00119-f002:**
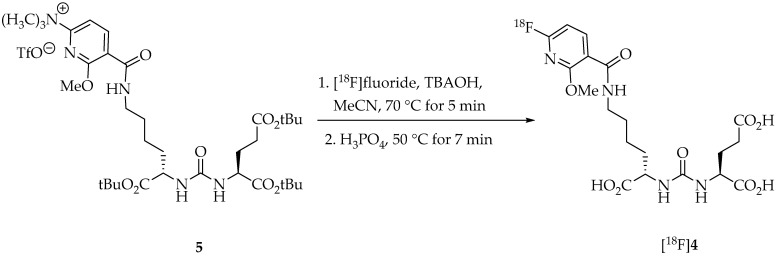
Synthesis of [^18^F]JK-PSMA-7 ([^18^F]**4**) by direct (“late-stage”) radiofluorination.

**Figure 3 pharmaceuticals-18-00119-f003:**
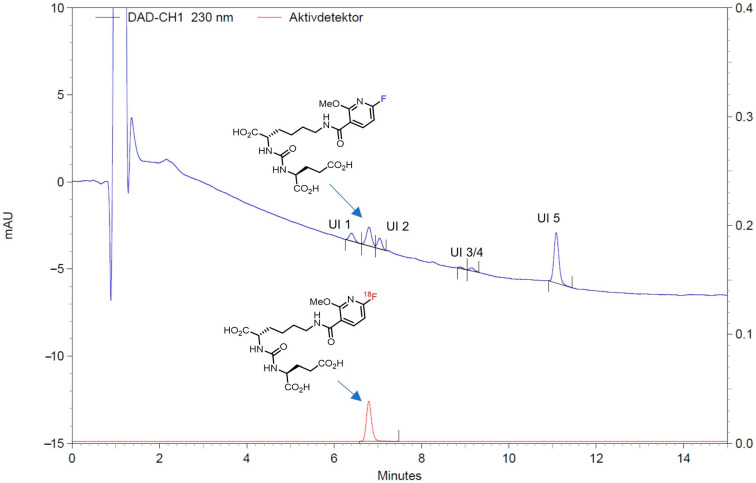
Example of a typical analytical HPLC chromatogram of the end product. Blue trace: UV channel; red trace: radioactivity. The amount of unknown UV impurities (UI 1–5) is below the limit of 5 µg/mL.

**Figure 4 pharmaceuticals-18-00119-f004:**
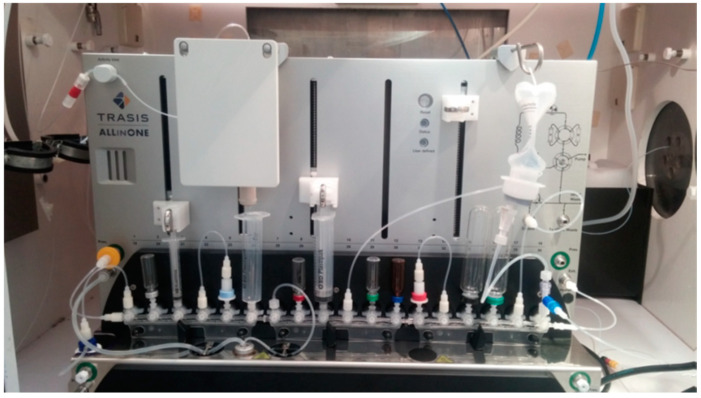
Trasis AllInOne synthesizer configured for the preparation of [^18^F]JK-PSMA-7.

**Figure 5 pharmaceuticals-18-00119-f005:**
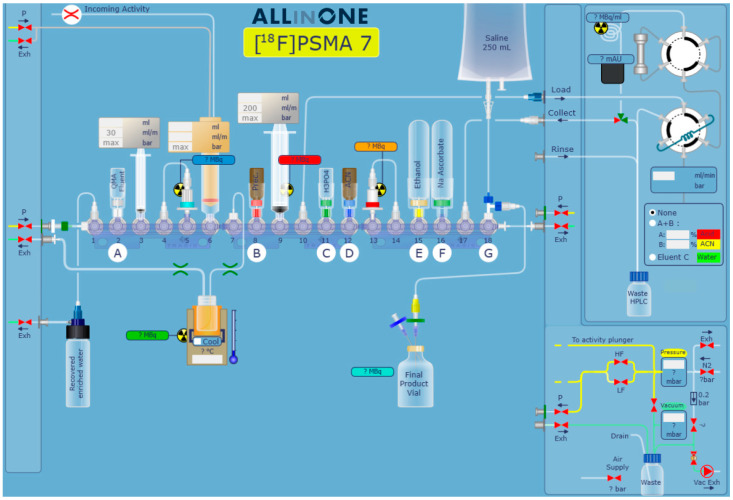
Flow diagram for synthesis of [^18^F]JK-PSMA-7 using the Trasis AllInOne synthesizer.

**Table 1 pharmaceuticals-18-00119-t001:** Tests, methods and acceptance criteria for the quality control of [^18^F]JK-PSMA-7.

Test	Method	Specifications at Release	Typical Results of a Production Batch (Here 24 August 2024)
*At product release*			
Appearance	Visual inspection	Clear and colorless, free from visible particles	Corresponds to specification
(Radio)chemical identity of [^18^F]JK-PSMA-7	HPLC (UV, γ-ray detectors)	[^18^F]JK-PSMA-7 peak in the radiochromatogram is consistent with the retention time of the reference standard to the nearest ± 0.5 min, taking detector delay into account	Corresponds to specification
Radionuclide identity of fluorine-18 (half-life measurement)	Dose calibrator	110 ± 5 min	109.6 min
Radiochemical identity of [^18^F]JK-PSMA-7	HPLC (γ-ray detector)	Retention time of [^18^F]JK-PSMA-7 consistent with retention time of JK-PSMA-7 (±0.5 min)	Corresponds to specification
Radiochemical purity of [^18^F]JK-PSMA-7	HPLC (γ-ray detector)	≥95% of total fluorine-18 activity	99.8%
pH	pH meter	4.5–8.5	6.31
Chemical concentration of JK-PSMA-7	HPLC (UV detector)	≤10 µg/mL	0.54 µg/mL
Sum of JK-PSMA-7 and all impurities above disregard limit	HPLC (UV detector)	≤5 µg/mL	4.65 µg/mL
Activity concentration	Dose calibrator	≥30 MBq/mL	1068 MBq/mL
Chemical concentration of tetrabutylammonium (TBA)	TLC spot test	<260 μg/mL	<260 μg/mL
Bacterial endotoxins	Kinetic chromogenic LAL method	≤17.5 IU/mL	<3.9 IU/mL
Sterile filter integrity	Pressure holding test	<20%	Corresponds to specification
Acetonitrile content	Gas chromatography	≤0.41 mg/mL	≤0.03 mg/mL
*After product decay*			
Radionuclide impurities with half-life > 2 h	γ-ray spectrometer	No signal higher than 5 times background noise	Corresponds to specification
Sterility	Direct inoculation	Sterile	Sterile

**Table 2 pharmaceuticals-18-00119-t002:** Assembly of the [^18^F]JK-PSMA-7 cassette on the Trasis AllinOne synthesizer, with description of reagent/material positions.

Manifold Position	Reagents or Materials Attached
1	PE tubing to [^18^O]water recovery vial
2	Eluents for QMA elution
3	Sterile 3 mL syringe with Luer Lock fitting
4	Silicone tubing connected to QMA cartridge in position 5
5	Preconditioned QMA cartridge
6	Sterile 20 mL syringe with Luer Lock fitting for collection of ^18^F-activity from cyclotron
7	Connector to 6 mL reactor with 2 tubes (male connector to cassette, female to exhaust)
8	Vial with precursor **5** (10 mg)
9	Sterile 20 mL syringe with Luer Lock fitting
10	PE tubing (OD 2 mm, length 150 mm) connected to Load port of the HPLC unit
11	Vial with phosphoric acid (1.1 mL)
12	Vial with MeCN (1 mL)
13	Sep-Pak C18 Plus cartridge
14	PE tubing (OD 2 mm) connected to Sep-Pak C18 Plus cartridge in position 13
15	Vial with EtOH (10.35 mL)
16	Vial with sodium ascorbate solution (10 mL)
17	PE tubing (OD 2 mm) connected to Collect port of the HPLC unit
18	Three-way valve connected to 0.9% NaCl for injection bottle and product vial

## Data Availability

The data are contained within the article.

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
