# Peer review of "Fully Automated Production of (((S)-1-Carboxy-5-(6-([18F]fluoro)-2-methoxynicotinamido)pentyl)carbamoyl)-l-glutamic Acid ([18F]JK-PSMA-7)â€"

_pharmaceuticals, 2025, doi:10.3390/ph18010119_

Round 1
Reviewer 1 Report
Comments and Suggestions for Authors
The manuscript entitled “Fully automated production of [18F]JK-PSMA-7” presents an automated method for the synthesis of the promising radiotracer [18F]JK-PSMA-7 using a one-pot protocol that complies with GMP guidelines. The manuscript demonstrates technical capability and is methodologically sound, but lacks significant novelty and does not offer a significant advance over the literatures. The manuscript effectively demonstrates a simplified one-pot synthesis of [18F]JK-PSMA-7. The automation of the cassette-based Trasis AllInOne system is well documented and consistent with GMP compliance requirements. This methodology will reduce the risk of cross-contamination, increase radiochemical yields, and improve reproducibility. However, despite the streamlined synthesis process, this work is an incremental improvement rather than a breakthrough in the production of radiotracers targeting PSMA. The novelty of this work is low, as several similar one-pot automated approaches for radiotracer synthesis have been reported in the literature (i.e., [18F]PSMA-1007). Furthermore, the statement made on line 43 is one that most radiochemistry experts would disagree with. Because most clinical trial radiotracers today tend to be purified using only SPE, excluding HPLC. In addition, the authors do not provide long-term production reproducibility results or at least results from several production runs. They do not provide adequate GMP-based quality control methods/information, and the results section contains a lot of information that should be described in Materials and Methods.
Taken together, the late-stage radiofluorination of onium triflate precursors, although effective, is not a novel or advanced chemistry approach. No significant breakthroughs in radiochemistry are seen, and this study is more appropriate for a specialized radiochemistry journal than for Pharmaceuticals.
Author Response
Comment: The manuscript entitled “Fully automated production of [18F]JK-PSMA-7” presents an automated method for the synthesis of the promising radiotracer [18F]JK-PSMA-7 using a one-pot protocol that complies with GMP guidelines. The manuscript demonstrates technical capability and is methodologically sound, but lacks significant novelty and does not offer a significant advance over the literatures. The manuscript effectively demonstrates a simplified one-pot synthesis of [18F]JK-PSMA-7. The automation of the cassette-based Trasis AllInOne system is well documented and consistent with GMP compliance requirements. This methodology will reduce the risk of cross-contamination, increase radiochemical yields, and improve reproducibility. However, despite the streamlined synthesis process, this work is an incremental improvement rather than a breakthrough in the production of radiotracers targeting PSMA. The novelty of this work is low, as several similar one-pot automated approaches for radiotracer synthesis have been reported in the literature (i.e., [18F]PSMA-1007). Furthermore, the statement made on line 43 is one that most radiochemistry experts would disagree with. Because most clinical trial radiotracers today tend to be purified using only SPE, excluding HPLC. In addition, the authors do not provide long-term production reproducibility results or at least results from several production runs. They do not provide adequate GMP-based quality control methods/information, and the results section contains a lot of information that should be described in Materials and Methods.
Taken together, the late-stage radiofluorination of onium triflate precursors, although effective, is not a novel or advanced chemistry approach. No significant breakthroughs in radiochemistry are seen, and this study is more appropriate for a specialized radiochemistry journal than for Pharmaceuticals.
Response by the authors:
We appreciate the reviewer’s thoughtful feedback and the opportunity to address the concerns raised. Below, we respond to each point in detail.
- Significance and novelty:
The manuscript was specifically prepared for the special issue in Pharmaceuticals entitled “Past, Present and Future Radiotracer Techniques: Radiopharmaceuticals in Cancer Theranostics”. A significant aspect of this topic is the GMP-compliant production of radiotracers for diagnostic applications. While one-pot synthesis and cassette-based automation are established concepts, the focus of our work is to address the practical challenges in implementing GMP-compliant radiopharmaceutical production processes. This includes overcoming issues of cross-contamination and ensuring reproducibility, both of which are critical for clinical translation and large-scale production.
While similar automated methods have been reported, such as for [18F]PSMA-1007, these typically involve multi-step processes or require multiple precursors, which introduce additional complexities for GMP validation and production. In contrast, the method described here employs a single-step late-stage radiofluorination approach, simplifying regulatory compliance and operational implementation. This streamlined process, as demonstrated, reliably produces [18F]JK-PSMA-7 with high radiochemical yield and reproducibility, which we believe adds practical value to the field.
- SPE vs. HPLC purification
We respectfully disagree with the statement that SPE purification is universally preferable for GMP-compliant syntheses. While SPE methods are commonly used, our experience has shown that they often carry a higher risk of non-radioactive impurities entering the final formulation. These impurities, while typically present at low concentrations, can exceed regulatory limits or require extensive analytical testing to detect and quantify.
HPLC purification, by contrast, provides a more robust and reliable method for achieving high-purity formulations, particularly for clinical-grade radiopharmaceuticals. We have therefore adopted HPLC purification for all GMP-compliant radiopharmaceutical syntheses, including [18F]JK-PSMA-7, to minimize the risk of impurities and ensure compliance with stringent quality control standards.
Revised text in the manuscript:
“The product was then finally purified by solid-phase extraction. To automate this two-step process using two precursors with an intermediate purification step, a device with two reactors was required. The GE TRACERLabTM FX2N synthesizer was selected for this purpose. To improve the purity of the product, a final HPLC purification also had to be integrated into the process (see supporting information for [22] for details).”
- Comparison with [18F]PSMA-1007
The radiosynthesis of [18F]PSMA-1007, as noted by the reviewer, involves a two-step process that can also be adapted to cassette-based systems. However, it requires the validation of two precursors for GMP suitability, which increases the regulatory and operational burden. In contrast, the direct one-step labeling process presented in our manuscript requires validation for only one precursor, offering significant advantages in terms of simplicity and compliance.
Revised text in the manuscript:
“The entire process took about 80 minutes and was prone to errors due to the many intermediate cleaning steps. Furthermore, two labeling precursors used under GMP conditions , as also used in the [18F]PSMA-1007 synthesis [20],must be considered for clinical application. Therefore, the process was reduced to a simplified two-step, one-pot synthesis and a cassette system, the Trasis AllinOne [38], was chosen as the synthesizer.”
- Long-term reproducibility
While the results presented in the manuscript are based on “only” 260 syntheses, the method has also been successfully implemented at other production sites. These sites report consistent and reliable production of [18F]JK-PSMA-7, further demonstrating the robustness and scalability of the process.
- Alignment with the journal’s audience
Given the increasing interest in [18F]JK-PSMA-7 as a diagnostic agent, we believe this manuscript will be of significant interest to the readership of this special issue. To further emphasize this aspect, we have included additional references in the revised manuscript that underscore the importance of [18F]JK-PMSA-7 for prostate cancer diagnostics. In addition, the automation methodology described here not only ensures compliance with GMP guidelines but also represents an important practical advancement in the production of radiopharmaceuticals targeting PSMA. Furthermore, the simplification and reliability of the presented process align with the goals of translating radiopharmaceuticals from the laboratory to routine clinical practice.
Reviewer 2 Report
Comments and Suggestions for Authors
The paper clearly describes fully automated synthesis of novel PSMA-selective tracer, the 18F-JK-PSMA-7, using Trasis all-in-one module. The paper is quite well written and I have just a few comments and suggestions to the authors:
1. Page 1, line 39 - While automation offers significant advantages, including consistency, repeatability and reduced human error, it may not always be mandatory. I suggest tempering the statement to acknowledge that specific contexts or needs might justify manual or semi-automated approaches without compromising quality.
2. Page 3, line 73 - You mention the use of phosphoric acid for final compound deprotection. Did you analyze possible trace (poly)phosphates in the final product (Table 3)? What about the triflate ion from the 10 mg of the precursor triflate salt?
3. Page 4, line 106, part 3.1. - Please add typical beam intensity and irradiation time in connection with the mentioned 18F activities.
Author Response
Comment 1: The paper clearly describes fully automated synthesis of novel PSMA-selective tracer, the 18F-JK-PSMA-7, using Trasis all-in-one module. The paper is quite well written and I have just a few comments and suggestions to the authors:
- Page 1, line 39 - While automation offers significant advantages, including consistency, repeatability and reduced human error, it may not always be mandatory. I suggest tempering the statement to acknowledge that specific contexts or needs might justify manual or semi-automated approaches without compromising quality.
Response by the authors:
We thank the reviewer for this insightful comment. We agree that automation, while advantageous, may not always be mandatory, and high-quality products can indeed be produced using manual or semi-automated systems in specific contexts. We have revised the sentence to reflect this perspective.
Revised text in the manuscript:
“In order to obtain sufficiently large activity amounts for medical use and to meet the regulatory requirements of “Good Manufacturing Practice” (GMP) for radiopharmaceuticals [26-28], automation of the production process is essential [29, 30]. Although high-quality products meeting GMP standards can also be produced using conventional, manually operated systems [31], these systems introduce additional challenges for regulatory compliance, such as the need for cleaning validation according to EU GMP Annex 15: Qualification and Validation. In contrast, cassette-based automated systems minimize the risk of cross-contamination, streamline GMP compliance, and enable more efficient production of radiopharmaceuticals like [18F]JK-PSMA-7.”
Comment 2: Page 3, line 73 - You mention the use of phosphoric acid for final compound deprotection. Did you analyze possible trace (poly)phosphates in the final product (Table 3)? What about the triflate ion from the 10 mg of the precursor triflate salt?
Response by the authors:
We thank the reviewer for bringing up this important point. While we have not specifically analyzed the potential presence of trace (poly)phosphates or triflate ions in the final product, we assume that these anions are effectively removed during the final reverse-phase HPLC purification step. However, we will incorporate this consideration into the upcoming drug approval process.
Comment 3. Page 4, line 106, part 3.1. - Please add typical beam intensity and irradiation time in connection with the mentioned 18F activities.
Response by the authors:
We thank the reviewer for this valuable suggestion. We have updated the manuscript to include the beam intensity (20 µA) and the irradiation time (1 -1.5 h).
Specific Comments
“Fluorine-18 was produced by irradiation (1 -1.5 h) of 18O-enriched water with 16.5 MeV protons and a beam current of about 20 µA using the 18O(p,n)18F nuclear reaction. Irradiations were performed at the baby cyclotron BC1710 (Japan-Steel-Works) at the Forschungszentrum Jülich using a titanium target.”
Reviewer 3 Report
Comments and Suggestions for Authors
This manuscript introduces an enhanced method for the automated synthesis of [18F]JK-PSMA-7, a radiotracer used in prostate cancer imaging. The proposed method represents a significant advancement in radiopharmaceutical production, offering notable improvements in yield, automation, and compliance with GMP.
Specific Comments
- Including a more detailed comparison with the previously established two-step synthesis method would be beneficial.
- A discussion on the storage conditions and stability of the final product would add value.
- A brief mention of any potential limitations or challenges of the new synthesis approach would be useful.
Author Response
Comment: This manuscript introduces an enhanced method for the automated synthesis of [18F]JK-PSMA-7, a radiotracer used in prostate cancer imaging. The proposed method represents a significant advancement in radiopharmaceutical production, offering notable improvements in yield, automation, and compliance with GMP.
Specific Comments
Comment 1: Including a more detailed comparison with the previously established two-step synthesis method would be beneficial.
Response by the authors:
We thank the reviewer for this valuable suggestion. In response, we have included a more detailed comparison of the two approaches in the manuscript.
Text added to the manuscript:
The originally developed synthesis of [18F]-JK-PSMA-7 [18F]4 was a two-step build-up synthesis that required two labeling precursors [22]. The 18F-building block, (6-[18F]fluoro-2-methoxypyridin-3-yl)-(2,3,5,6-tetrafluorophenyl)methanone ([18F]2), was synthesized by nucleophilic 18F-fluorination of 6-methoxy-N,N,N-trimethyl-5-(2,3,5,6-tetrafluorobenzoyl)pyridin-2-aminium triflate (1) under the conditions of a modified minimalist approach [37] to give the desired intermediate [18F]2. The [18F]fluoride was previously fixed on a QMA cartridge, washed with anhydrous acetonitrile and the [18F]fluoride eluted with a solution of the precursor 1 in acetonitrile-MeCN, followed by a mixture of acetonitrile-MeCN and tBuOH and heated. This intermediate then had to be purified by solid-phase extraction before it was coupled to the urea derivative Lys-C(O)-Glu 3. To do this, the active ester [18F]2 was eluted in a vessel with Lys-C(O)-Glu 3 and Et4NHCO3 in anhydrous ethanol. After stirring for 3–5 minutes, the protective groups were hydrolyzed with a 0.1% TFA solution. The product was then finally purified by solid-phase extraction. To automate this two-step process using two precursors with an intermediate purification step, a device with two reactors was required. The GE TRACERLabTM FX2N synthesizer was selected for this purpose. To improve the purity of the product, a final HPLC purification also had to be integrated into the process (see supporting information for [22] for details). To remove the acetonitrile used as an HPLC solvent, the eluent containing the product was diluted with water and fixed on a further solid phase cartridge. The fixed product cartridge was then washed with water and the product eluted with EtOH in isotonic saline diluted with isotonic saline to reduce the alcohol content to 10%. The entire process took about 80 minutes and was prone to errors due to the many intermediate cleaning steps. Furthermore, two labeling precursors used under GMP conditions must be considered for clinical application. Therefore, the process was reduced to a simplified two-step, one-pot synthesis and a cassette system, the Trasis AllinOne [, was chosen as the synthesizer.
Comment 2: A discussion on the storage conditions and stability of the final product would add value.
Response by the authors:
We thank the reviewer for this valuable suggestion. Stability measurements were indeed conducted as part of the synthesis validation process. The results have now been incorporated into the manuscript to provide a more comprehensive discussion on the stability of the final product.
Text added to the manuscript:
“As part of the validation of the synthesis, the stability of the [18F]JK-PSMA-7 injection solution was also examined over a period of 8 hours. No significant decomposition was detected during this time frame, demonstrating that the product is stable and remains suitable for use within this period.”
Comment 3: A brief mention of any potential limitations or challenges of the new synthesis approach would be useful.
Response by the authors:
We thank the reviewer for this insightful suggestion. In response, we have added a brief discussion on the potential limitations and challenges of the new synthesis approach at the end of the conclusion section.
Text added to the conclusion:
“However, the adoption of cassette-based systems assumes access to this specialized equipment, which may not be available in all facilities. Nevertheless, the increased synthesis reliability and improved GMP compatibility makes them a worthwhile choice for the production of [18F]JK-PSMA-7 as well as other radiopharmaceuticals. Another potential limitation of the synthesis lies in the use of higher starting activities of fluorine-18. In this study, no radiolysis issues were observed with the 40 GBq starting activity used. Nevertheless, to mitigate potential risks, the formulation was carried out with sodium ascorbate as a radical scavenger. This precaution ensures that syntheses with starting activities of up to 100 GBq could be performed without radiolysis-related complications. However, for even higher starting activities, the risk of radiolysis cannot be entirely excluded and would require further investigation.”
Round 2
Reviewer 1 Report
Comments and Suggestions for Authors
Thank you for your thorough and well-reasoned responses to my previous comments. I appreciate the effort and logical arguments you have made in addressing the points I raised. Your emphasis on GMP-compliant production, the streamlined one-pot synthesis, and the advantages of HPLC purification over SPE are duly noted and acknowledged as valid contributions to radiopharmaceutical production processes.
However, I must respectfully maintain my stance on the novelty and significance of your manuscript. As you may be aware, TRASIS company currently offers GMP-grade cassettes for their All-in-One automated synthesizer (also used in your study), which are capable of producing [18F]PSMA-1007 in approximately 40–50 minutes, achieving radioactivity yields of 40–60%. Similarly, GMP-grade synthesis of [18F]JK-PSMA-7 using All-in-One systems has already been demonstrated and commercialized with more shorter timeframes (~55 min) than reported in your study (~80 min) while maintaining comparable radioactivity yields. These existing processes highlight the critical importance of synthesis time in radiochemical workflows and their well-established utility in GMP-compliant environments.
While your work contributes incremental improvements in the simplification and GMP compliance of [18F]JK-PSMA-7 production, these advancements are, in my view, not sufficiently novel or groundbreaking to warrant publication in pharmaceuticals. I believe this manuscript would be more appropriately suited for journals such as diagnostics, chemistry (MDPI) or specialized radiochemistry journals, where the technical and incremental contributions would be of greater interest to the readership.
I appreciate your dedication to refining this manuscript and the valuable discussions it has generated.